# RIF1 Links Replication Timing with Fork Reactivation and DNA Double-Strand Break Repair

**DOI:** 10.3390/ijms222111440

**Published:** 2021-10-23

**Authors:** Janusz Blasiak, Joanna Szczepańska, Anna Sobczuk, Michal Fila, Elzbieta Pawlowska

**Affiliations:** 1Department of Molecular Genetics, Faculty of Biology and Environmental Protection, University of Lodz, Pomorska 141/143, 90-236 Lodz, Poland; 2Department of Pediatric Dentistry, Medical University of Lodz, 92-216 Lodz, Poland; joanna.szczepanska@umed.lodz.pl; 3Department of Gynaecology and Obstetrics, Medical University of Lodz, 93-338 Lodz, Poland; anna.sobczuk@umed.lodz.pl; 4Department of Developmental Neurology and Epileptology, Polish Mother’s Memorial Hospital Research Institute, 93-338 Lodz, Poland; michal.fila@iczmp.edu.pl; 5Department of Orthodontics, Medical University of Lodz, 92-217 Lodz, Poland; elzbieta.pawlowska@umed.lodz.pl

**Keywords:** RIF1, replication timing, DNA double-strand break repair, reactivation of replication fork, TP53BP1, BRCA1

## Abstract

Replication timing (RT) is a cellular program to coordinate initiation of DNA replication in all origins within the genome. RIF1 (replication timing regulatory factor 1) is a master regulator of RT in human cells. This role of RIF1 is associated with binding G4-quadruplexes and changes in 3D chromatin that may suppress origin activation over a long distance. Many effects of RIF1 in fork reactivation and DNA double-strand (DSB) repair (DSBR) are underlined by its interaction with TP53BP1 (tumor protein p53 binding protein). In G1, RIF1 acts antagonistically to BRCA1 (BRCA1 DNA repair associated), suppressing end resection and homologous recombination repair (HRR) and promoting non-homologous end joining (NHEJ), contributing to DSBR pathway choice. RIF1 is an important element of intra-S-checkpoints to recover damaged replication fork with the involvement of HRR. High-resolution microscopic studies show that RIF1 cooperates with TP53BP1 to preserve 3D structure and epigenetic markers of genomic loci disrupted by DSBs. Apart from TP53BP1, RIF1 interact with many other proteins, including proteins involved in DNA damage response, cell cycle regulation, and chromatin remodeling. As impaired RT, DSBR and fork reactivation are associated with genomic instability, a hallmark of malignant transformation, RIF1 has a diagnostic, prognostic, and therapeutic potential in cancer. Further studies may reveal other aspects of common regulation of RT, DSBR, and fork reactivation by RIF1.

## 1. Introduction

Replication timing (RT) is a cellular program to coordinate DNA replication initiations in different chromosomal regions (reviewed in [1]). This program ensures that all chromosomes are replicated once and only once per cell cycle. Although “timing” suggests that RT regulates DNA replication temporally, RT is currently understood as a program-controlling replicative DNA synthesis both temporally and spatially [2].

DNA double-strand breaks (DSBs) are one of the most serious DNA damages, leading to cell death, cancer transformation, and other danger consequences if not repaired or misrepaired. The process of DNA replication is inherently associated with alterations in topological relationships within DNA that may cause replication fork stalling. The restoration of fork activity may produce DSBs as an intermediate that must be repaired to resume DNA replication (reviewed in [3]). In general, three DSBs repair (DSBR) pathways are considered: homologous recombination repair (HRR), non-homologous end joining (NHEJ), and single-strand annealing (SSA), with several alternative pathways of HRR and NHEJ (reviewed in [4]).

The choice between DSBR pathways is not completely clear and it depends on the nature of a DSB and its cellular context (reviewed in [5]). Effective DSBR is important in resuming disturbed DNA replication so the fast and proper choice of DSBR mode is essential in cells replicating their DNA. It seems that during DNA replication, HRR should be preferred as a DSBR mode because chromosomes in S phase are extended, facilitating finding a homology partner, which usually is sister chromatid. Moreover, HRR is less error-prone than the remaining two DSBR pathways, which is of a special significance in DNA replication.

It is clear that impaired DNA replication from a single origin impairs RT, if not resumed in an appropriate time. This resuming may be associated with the generation of DSBs as an intermediate of fork repair process. These DSBs must be repaired to preserve replication and RT. Therefore, there are functional interconnections between RT, DSBR, and fork reactivation.

Replication timing regulatory factor 1 (RIF1, RAP1-interacting factor 1) is a key regulator of RT and plays a role in DSBR choice, so it is a candidate to adjust DSBR to RT [6,7]. Apart from these activities, RIF1 can be involved in several other processes, including nuclear organization, regulation of telomerase and telomere maintenance, transcriptional regulation of gene clusters, regulation of stem cell pluripotency, and resolution of ultrafine DNA bridges in anaphase, but the mechanisms underlying these involvements are not completely clear [8,9,10,11]. Although there are excellent reviews on some specific activities of RIF1, there is not a comprehensive, in-depth paper presenting RIF1 and its involvement in RT, DSBR, and fork reactivation [12,13,14,15,16]. In this review, we present and update information on RIF1 structure and functions in RT, replication fork reactivation, and choice and mechanisms of DSBR. Furthermore, we provide some data and arguments supporting the hypothesis that RIF1 may regulate RT and DSBR in mutually related mechanisms. We tried to focus on the mammalian, especially human, RIF1, but due to the still incomplete knowledge on RIF1 functions, sometimes data obtained in yeast are provided.

## 2. RIF1—The Gene and the Protein

Rif1 was discovered in yeast as a telomere-binding protein [17]. However, human RIF1 does not bind functional, but dysfunctional telomeres in cells maintaining telomeric DNA in the absence of telomerase by alternative lengthening of telomeres [18]. Such dysfunction-related localization is typical for DNA damage response (DDR) factors [19].

The human *RIF1* gene has 41 exons and is located at 2q23.3 in plus strand and its most recent chromosomal allocation in the human genome is 151,409,883–151,534,200 (GRCh38/hg38) (https://www.ncbi.nlm.nih.gov/gene/55183, accessed on 5 October 2021). The human *RIF1* gene is conserved in chimpanzee, Rhesus monkey, dog, cow, mouse, rat, and chicken, and hundreds of organisms have orthologs with human *RIF1*. It has six mRNA splice variants and six protein isoforms. The maximal expression of the *RIF1* gene is observed in late G2/S phase, suggesting that then RIF1 may perform its most important functions in this phase of the cell cycle [20].

The human RIF1 protein (recommended name: telomere-associated protein RIF1) has 2472 aa and molecular weight of 274,466 kDa. It is a *trans*-acting factor regulating RT in humans (reviewed in [2]). Three domains can be distinguished in the human RIF1: N-terminal domain (NTD), C-terminal domain (CTD) and the intrinsically disordered polypeptide (IDP) domain. NTD contains over a dozen of HEAT (huntingtin, elongation factor 3, a subunit of protein phosphatase 2A and TOR)/Armadillo-like helical repeats) [21] (Figure 1). The IDP domain spanning between NTD and CTD has several phosphorylation sites and may be important for DNA binding by RIF1 [22,23]. Wang et al. showed that phosphorylation of yeast Rif1 had both positive and negative effects on telomere length regulation [24]. These authors suggested that as all yeast Rif 1 orthologues, including human one, had one or more conserved serine or threonine followed by glutamine (SQ/TQ) cluster domains targeted by the ATM family kinases, the results they obtained might have a more general significance and implications for the regulation of RIF1 functions in human and other organisms. RIF1 CTD may directly interact with DNA and display a higher affinity to double-stranded DNA (dsDNA) than single-stranded DNA (ssDNA). NTD and CTD of human RIF1 form oligomers that bind DNA [22,23]. Mouse RIF1 CTD contains three CI-CIII segments, among them CII is partially folded and can bind DNA [21,25]. RIF1 also has the SILK domain, which is PP1 (protein phosphatase 1) interacting motif located in humans close to the IDP/CTD boundary [26]. As shown in subsequent sections, this domain has a functional significance. The highly conserved residues of RVxF are an essential docking motif for PP1 binding. Human RIF1 has BLM interacting domain, which supports recruiting the BLM helicase to stalled replication fork. Structural features of RIF1 and its ability to oligomerize underline its preferential binding to G4-quadruplexes (G4s) DNA, a four-stranded DNA resulting from the ability of guanine bases to form stable hydrogen bonds with other guanines [27,28]. In human cells, RIF1 localizes at nuclease-insoluble structures at the nuclear membranes and the periphery of the nucleoli. More details about the structure of the RIF1 protein were established in yeast and can be found elsewhere (reviewed in [29]).

Apart from the involvement in RT and DSBR, RIF1 performs several other vital cellular functions. DSBR is not the only role RIF1 may play in the genome maintenance and DDR. It may protect newly replicated DNA from degradation when the replication fork stalls [30,31]. It is claimed that the fork-reactivating functions of RIF1 are independent of its DSBR repair activity, but this statement is somehow controversial and further studies are needed to deeper explore this issue [31]. RIF1 plays a role in embryonic stem cells (ESCs) development, directly repressing activation of endogenous retrovirus region with the involvement of the HEAT domain [32]. An important role of RIF1 in ESCs was confirmed in other research showing that RIF1 inactivation in mice was mostly lethal and few survivors showed developmental abnormalities, including growth retardation [33]. This effect was explained by the S phase functions of RIF1 as presented in more detail in subsequent sections.

As DNA repair is critical for genomic stability and cancer transformation, RIF1 can be considered as an anti-apoptotic factor and target in anti-cancer treatment. Wang showed a high expression of RIF1 in human breast carcinoma cells positively correlated with differentiation status of invasive ductal breast tumors [34]. RIF1 may promote cancer progression by binding to the promoter of the human telomerase reverse transcriptase gene encoding the catalytic subunit of telomerase, an enzyme essential for cancer transformation in many tumors [35]. Another cancer-related mechanism of RIF1 is the activation of the Wnt/β-catenin signaling in non-small-cell lung carcinoma mediated by PP1 [36,37]. This suggests RIF1 potential as a transcription modulating factor. *RIF1* mutations were detected in human breast cancer lines [38,39].

## 3. Replication Timing

A eukaryotic nuclear chromosome, contrary to its bacterial counterpart, contains many regions of the initiation of DNA replication. Therefore, the activation (firing) of these regions should occur in a spatially and temporally ordered fashion to ensure complete duplication of the genome, avoid structural disturbances, and spare energy. Moreover, DNA replication should be adjusted to other ongoing processes within a cell. Replication timing coordinates the initiation of replication in different regions of a chromosome and is primarily responsible for the structural organization of the genome within the chromatin (reviewed in [40,41]).

Eukaryotic chromosomes need to be replicated once and only once, which is a challenge for the cell due to a high number of origins of replication. In the late mitosis and early G1 phase, origins in long, up to several megabases in length, genome segments are “licensed” by loading the double hexameric MCM2-7 (minichromosome maintenance complex components 2–7, MCM) complexes (reviewed in [42]). Licensed origins stay inactive until the beginning of S phase. Several proteins facilitate recruiting and loading of MCM, including CDC6 (cell division cycle 6), CDT1 (chromatin licensing and DNA replication factor 1) and the origin recognition complex (ORC). Altogether, these proteins form the pre-replicative complex (pre-RC) [43].

Next, the interaction among several proteins, including TOPBP1 (DNA topoisomerase II binding protein 1), TICCR (TOPBP1 interacting checkpoint and replication regulator, Treslin), RECQL4 (RecQ like helicase 4), MTBP (MDM2 binding protein), and others, causes recruiting of CDC45 (cell division cycle 45) and the four-subunit GINS complex consisting of Sld5 (GINS complex subunit 4), Psf1 (GINS complex subunit 1), Psf2, and Psf3, to MCM to form the preinitiation complex (pre-IC). The pre-IC formation occurs at the G1-S transition and is mediated by CDKs (cyclin dependent kinases) and DBF4 (DBF4 zinc finger)-dependent kinase (DDK). Phosphorylation of MCM by DDK and other factors of the pre-IS complex by CDKs induces activation of the CMG (CDC45-MCM-GINS) complex, which is the human replicative helicase, local DNA denaturation, and recruitment of other replication factors, including a replicative DNA polymerase (replicase) to form functional replisomes to mediate bidirectional DNA synthesis [44]. Therefore, licensing, pre-IC formation, and origin firing are three crucial steps in the initiation of DNA replication.

In general, more origins are licensed than required in S phase and these extra origins are used to replicate unreplicated DNA fragments resulting from replicative stress causing stalling of the replication fork [45,46].

The positive regulation of the initiation of DNA replication is counteracted by a negative control of this process [47] (reviewed in [48]). In humans, excessive licensing in G1 is inhibited by chromatin compaction through methylation of Lys 20 in histone H4 mediated by KMT5A (lysine methyltransferase 5A) [49].

Replication timing is mediated by the generation of genome segments that contain origins fired in the specific times of S phase [50] (Figure 2). Such a timing of replication initiation spares energy expedited by the cell as some regions of the genome need not to be actively fired. Furthermore, timing of initiation decreases a chance for a steric conflict between multi-protein initiation complexes. The RT program is established shortly after mitosis at the timing decision point (TDP) in G1 [51]. Not all pre-RCs become active replication origins. At the origin decision point (ODP) in G1, certain pre-RCs are selected to become DNA replication initiators, while others remain inactive during S phase [52].

Many earlier works suggested that different origins within a eukaryotic chromosome are activated in early, middle, or late S phase. The first RT genome-wide experiment was performed in budding yeast and was methodologically like the classical Meselsohn-Stahl experiment [53]. Two main conclusions of that study were that (i) times of firing origins belong to a continuous rather than discrete (early-middle-late) spectrum, and (ii) replication was earlier in chromosomal regions located near centromeres than in their telomere counterpart. Early replication appears in the regions that are rich in transcriptionally active genes and late replication is linked with gene-poor regions.

Most regions of the genome replicate both alleles synchronously, but some loci do so in an asynchronous way, with one allele replicating in early and the other in late S [54,55]. Genes with biallelic expression replicate synchronously, while genes that are mono-allelically expressed, including imprinted genes, allelically excluded genes, and genes on female X chromosomes, replicate asynchronously (reviewed in [56]). However, some genes with biallelic expression show asynchronous replication mode, with the maternal allele replicating early in some population of cells, while in others it is the paternal allele that replicates first [57]. Recently, Blumenfield et al. revealed a highly coordinated, RT-based program to mark regions in the human genome to express in an allele-differential way, contributing to many gene functions and cell identities [58]. This program of allele-specific RT is established in early embryo and maintained in all descendant generations of somatic cells.

Replication timing is characterized by high conservation within a tissue; the variation in cell type-specific RT among individuals is usually not greater than 2% [59]. However, there are much higher variations in RT for different cell types or within a specific cell type during development [60]. Therefore, RT can be considered as a marker of cell type and stage of development. This provokes a question about RT in various pathological states, particularly in cancer, as cancer cells have their own replication program, different from their normal counterparts. Deregulation of RT in cancer has been reported in several studies, but it is still not known whether disturbances in RT belong to reasons or consequences of cancer transformation (reviewed in [61]).

Timing regulation of replication origin firing is underlined by several mechanisms. Origins that are to be fired early may be marked by specific compounds in addition to pre-RC complexes. In yeast, Fkh1 (fork head homolog 1) marks pre-RC complexes to be fired early by the interaction with CDC7 and facilitates binding CDC45 by these origins and drives their relocation from nuclear periphery to the interior [62]. The open structure of chromatin supports early firing. The availability of initiation factors naturally limits the number of origins that may be fired early and selects the remaining origins to be fired late [63]. Alterations in replication timing are mutually dependent on the structure of the genome, including its sequence and epigenetic pattern (reviewed in [64]). RIF1 prevents middle/late origin firing by mechanisms that are presented in subsequent sections [13].

## 4. RIF1 in Replication Timing

Regulation of RT is the best recognized and most important function of RIF1; however, it is not exactly known how the remaining RIF1 activities can be related to its main functions.

As mentioned, the initiation of DNA replication requires activities of DDK targeting the MCM2-7 proteins. Alver et al. showed RIF1-mediated MCM dephosphorylation at replication fork [65]. RIF1 guided PP1 to chromatin to invert DDK-mediated phosphorylation of MCM, i.e., to inhibit initiation of DNA replication (Figure 3). RIF1 depletion increased phosphorylation of MCM and the rate of origin firing. In conclusion, replication initiation and replisome stability depended on the state of MCM phosphorylation, which was maintained by the balance between phosphorylation and dephosphorylation of MCM at replication fork, mediated by DDK and RIF1, respectively.

### 4.1. Organizing the 3D Structure of Chromatin

Although the replication domain model underlies the mutual dependence of temporal and spatial regulations of replication initiation, such dependence is complex [66]. Temporal regulation of replication initiation is only possible when all origins are spatially accessible for proteins involved in the initiation of DNA replication. Therefore, the structure of chromatin is crucial for coordinated initiations of DNA replication [40,41].

Klein et al. showed that loss of RIF1 resulted in almost complete elimination of the RT program by enhancing RT heterogeneity between individual cells, widespread alterations in chromatin modifications, and genome compartmentalization in various human cell lines [67] (Figure 4). These authors confirmed that RT played a key role in the maintenance of the cellular epigenetic profile. Therefore, RIF1 is essential for RT and the mechanism of its action can be underlined by the induction of structural changes in chromatin. This important work showed that RIF1 was a key regulator of the epigenetic pattern through its involvement in the RT control.

Ryba et al. confirmed the presence and the human-mouse conservation of coordinately replicated domains interspersed by origin-suppressed regions in four hESC lines, hESC-derived neural precursor cells (NPCs), lymphoblastoid cells, and human-induced pluripotent stem cell lines (hiPSCs), in comparison with related mouse cell types [68]. Changes in RT induced by differentiation of these cells were coordinated with transcriptional alterations. The presence of topological domains (topologically associated domains, TADs) is a pervasive structural feature of the genome organization that is conserved across species [69]. Initial research suggested that the TADs borders overlapped with the borders of coordinately replicating regions [68]. However, subsequent studies suggested that RT domains might be differently organized than TADs. Both cohesin or its loading factor NIPBL (NIPBL cohesin loading factor) and the CCCTC-binding factor disordered TADs, with no effect on RT domains [70,71,72]. Hi-C (high-through chromosome conformation capture) compartments A/B were better correlated with early/late replication domains than TADs [41,73].

Foti et al. showed that RIF1, assisted by lamin B1, coated late-replicating domains in mouse embryonic stem cells [74] (Figure 5). RIF1 was an essential determinant of replicating late domains that were not bound by lamin B1. These authors also showed that RIF1 controlled the interaction between RT domains during G1 and S phases, suggesting that RIF1 linked nuclear architecture and RT programming, organizing correct 3D compartments.

Several other experiments, including those with soluble and insoluble nuclear structures, suggest that RIF1 plays an important role in the control of higher-order chromatin structure (reviewed in [75]).

As global spatial organization of chromatin plays a role in RT functioning, the question of whether such a role can be also attributed to 3D structure of DNA is justified. Strikingly, RIF1 displays a high affinity to G4 DNA, so the next question arises how this affinity may influence the role of RIF1 in RT regulation (reviewed in [29]).

### 4.2. Interaction with G4-Quadruplexes

Guanine has this unique property that it can form stable hydrogen bonds with two other guanines, which can feature the formation of guanine quartets (G4s), plying an emerging role in biology [76]. Therefore, if a single strand of DNA is guanine rich, which is the case of telomeric DNA, it can fold onto itself, forming a G4 quadruplex, a four-stranded DNA, held together by hydrogen bonds within G4s. Similarly to dsDNA, some other weak electrostatic interactions may contribute to the stability of G4 quadruplexes. Two “independent” DNA duplexes may also form a G4 quadruplex, which can also be formed by RNAs and DNA–RNA hybrids.

G4 quadruplexes may regulate chromatin structure and associate with TAD boundaries [76]. Human telomeric DNA, which is guanine-rich and single-stranded, has a natural tendency to form a G4 quadruplex. As RIF1 binds telomeres, this binding may be associated with its G4 quadruplex preference.

Most works on G4 binding by RIF1 were performed in yeasts, but the G4 binding properties of yeast Rif1 are very similar to its mammalian counterpart [29]. Thirty-five binding sites for Rif1 were identified in fission yeast chromosomes [14]. These sites mostly localized near silent origins and contained at least two copies of the conserved sequence, CNWWGTGGGGG (N—any base, W—A/T), and base change in it caused the loss of RIF1 binding and activation of late-firing or dormant origins located even 50 kb away. These studies showed that RIF1-binding sites adopted G4-quadruplex-like structures in vitro.

N-terminal HEAT repeats and the non-identified part of CTD of RIF1 can independently bind G4 quadruplexes [22,23]. RIF1 binds chromatin at maximal rate in G1 and then slower in S phase [13,14]. Thanks to its highly oligomerized structure, RIF1 may bind several G4 quadruplexes at a time [27].

These and other works suggest that RIF1 recognized and bound G4-quadruplex-like structures and generated local chromatin structures that might suppress origin firing over a long distance. These structures are generated by condensation of chromatin fibers [77]. Therefore, the interaction with G4 quadruplexes may be essential for the regulation of RT by RIF1.

Genomic instability supports the fast accumulation of genetic and epigenetic changes fueling cancer transformation, and otherwise impairs normal cellular functioning [78]. Replication stress, a phenomenon causing stalling, breakage, or collapse of replication fork, affects RT (reviewed in [64,79]). Replication stress is targeted by DDR, the main collective mechanism to maintain genomic stability [80]. Complex DNA structures within the replication fork resulting from replication stress are resolved by DDR with various intermediates, including DSBs, one of the most serious DNA damages with a strong causal association with genomic instability and cancer (reviewed in [81] and discussed in the next sections). Clearly, DSBs induced by various endogenous and exogenous factors may impair RT as they belong to the most influencing cases of replication stress.

## 5. RIF1 in Double-Strand Break Processing

DNA double-strand breaks strongly contribute to genomic instability and are potentially lethal if not repaired or misrepaired. DSBs are primarily sensed by the MRN (MRX in yeast)–MRE11 (MRE11 homolog, double strand break repair nuclease)/RAD50 (RAD50 double strand break repair protein)–NBS1 (nibrin, Nijmegen breakage syndrome 1) complex (reviewed in [82]). MRN triggers DDR by the interaction with ataxia telangiectasia mutated (ATM, ATM serine/threonine kinase) and ATR (ATM-and-RAD3-related, ATR serine/threonine kinase) proteins, causing the DSB signaling, signal transduction, and effectors action, including DNA repair. As mentioned, in response to DSBs, the cell has evolved three general DSBR pathways: HRR, NHEJ, and SSA, with several variants to target DSBs (reviewed in [4]). The mechanisms of choice of DSBR depend on the structure of DNA broken ends, cell type, cell cycle phase/chromatin organization, and several other factors, and are not completely clear [83]. Accumulating evidence suggests that the choice of DSBR pathway in replicating cells differs from that in their non-replicating counterpart (reviewed in [5]). Here, we limit our considerations to HRR and NHEJ as they are much more common than SSA [84].

Although the choice between HRR and NHEJ is temporally regulated by the cell cycle progression, end resection, a prerequisite of HRR rarely occurring in NHEJ, may be a major operational factor deciding the choice of DSBR pathway at a DSB with blunt ends. Also, the structure of DSB-associated ends may decide the DSBR pathway. One-ended DSBs cannot be repaired by NHEJ or SSA as there is no other end to join. As estimated by Shibata, in G2 about 70% of all two-ended DSBs are repaired by NHEJ, and the remaining 30% by HRR [85].

Uncapped telomeres can be recognized as one-sided DSBs and proceeded by DDR resulting in fusion chromosomes, containing fusion genes that frequently act as oncogenes (reviewed in [86]). Therefore, DDR factors that constitutively serve to protect telomeres, may induce their fusion. The nucleotide excision repair endonuclease ERCC1 (XRCC1, X-ray repair cross complementing 1)/XPF (XPF, DNA damage recognition and repair factor) and NHEJ factors Ku70 (X-ray repair cross complementing 5)/80 and DNA-PKcs (protein kinase, DNA-activated, catalytic subunit) are examples of DDR components that may be involved in such ambiguous action [87,88].

It was shown that yeast Rif1 negatively regulated telomere length through its conserved HEAT repeats independently of its role in RT [89,90]. However, as underlined above, the role of human RIF1 in telomere maintenance is clearly distinct from that of yeast Rif1. Human RIF1 does not bind telomere capping complex and joins telomeres only if they are uncapped, broken, or critically short [20,91,92]. RIF1 binds at dysfunctional mammalian telomeres to recover them with its DSBR-related activity rather than telomere-specific functions. However, Adams and McLaren showed that mouse RIF1 highly expressed in primordial germ cells and embryo-derived pluripotent stem cell lines directly interacted with the telomere-associated protein TRF2 and could be crosslinked to telomeric repeat DNA in mouse embryonic stem cells [93]. These authors also showed that RIF1 assisted to limit the expression of the ZSCAN4 (zinc-finger and SCAN domain-containing 4) gene, whose product supports a recombination-related mechanism of telomere-elongation. Therefore, RIF1 can be involved in DSBR-independent telomere maintenance in the germline and early mouse development. Further works are needed to definitely determine the role of mammalian RIF1 in telomere maintenance and clarify the reason of difference between that role and its yeast counterpart.

TRF1 (TERF1, telomeric repeat binding factor 1) is a protein involved in telomere maintenance [94]. It was shown that a fraction of TRF1 may exist in a free form, not bound to telomeric chromatin, when phosphorylated at T371 by CDK1 [95]. McKerlie et al. showed that (pT371)TRF1 was recruited to DSB sites and formed DSB-induced foci in response to DSB-inducing factors [96]. They also showed that depletion of either TP53BP1 (53BP1, tumor protein P53 binding protein 1) or RIF1 stimulated the recruitment of (pT371)TRF1 to DSB sites. Therefore, RIF1 may, independently of TP53BP1, inhibit the binding of an important DDR protein.

### 5.1. Essential Role of TP53BP1

In most effects related to DSBR, RIF1 associates with TP53BP1 [97]. TP53BP1 is a chromatin-binding DDR protein that is recruited to DSB sites via the interaction of its Tudor domain with dimethylated lysine 20 of histone H4 (H4K20me2) [98]. TP53BP1 regulates the choice of DSBR pathway by inhibiting resection of DNA ends by nucleases and consequently suppressing HRR [99].

Several mechanisms may underline regulatory action of TP53BP1, including enforcing a chromatin barrier against nucleases [100]. CBP (CREB binding protein)-mediated acetylation of K1626/1628 in the ubiquitylation-dependent recruitment (UDR) motif was reported to impair the interaction of TP53BP1 with nucleosomes and consequently block the recruitment of TP53BP1 and its downstream factors RIF1 and PTIP (PAX interacting protein 1) to DSB sites [101,102]. The acetylation of TP53BP1 was tightly regulated by histone deacetylase 2 (HDAC2) to sustain the balance between the HRR and NHEJ pathways.

Silverman et al. observed that human RIF1 accumulated at DSB sites and formed foci colocalizing with DDR factors [18]. They did not observe an accumulation of RIF1 at functional telomeres. Formation of the DDR foci required the function of ATM, but not ATR. RIF1 involvement in the DDR foci was strongly dependent on TP53BP1. However, the RIF1 response was not related to several other important DDR proteins, including CHK2 (checkpoint kinase 2), MRE11, NBS1, and BRCA1 (BRCA1 DNA-associated repair). The intra-S checkpoints slowed down DNA synthesis to give the cell more time to repair DNA damage, and RIF1 functioned in one of the MRN-mediated and ATM-regulated intra-S-phase checkpoints. The authors concluded that RIF1 had a unique pattern of regulation at DSB sites depending primarily on ATM and TP53BP1.

Wang et al. demonstrated that RIF1 downregulation caused defective HRR in cancer cells making them more sensitive to DSB-inducing anticancer drugs [34]. This might result from HRR-mediated DSBR in pre-replicative DNA with no possibility to inhibit end resection by RIF1.

Grabarz et al. showed that the BLM (BLM RecQ like helicase) helicase suppressed long-range deletions generated by alternative end-joining [103]. BLM did so in an epistatic manner with TP53BP1 and RIF1. In turn, Zimmerman et al. showed that RIF1 impaired the resection factors CtIP (RB binding protein 8, endonuclease), BLM, and EXO1 (exonuclease 1), and limited accumulation of BRCA1-BARD1 (BRCA1 associated RING domain 1) complexes at DSB sites. [104]. These results confirm the role of RIF1 as an important element of the DSBR control by TP53BP1 [97].

Bakr et al. showed that RIF1 acted epistatically with TP53BP1 to prevent DNA end resection at DSBs mediated by BRCA1/CtIP [105]. They also showed that RIF1 depletion in G1 cells caused the accumulation of BRCA1, CtIP, and RPA (replication protein A), but not RAD51 foci. Interestingly, PARP1 (poly(ADP-ribose) polymerase 1)-dependent end joining (PARP1-EJ) was activated in RIF1-depleted G1 cells. Therefore, in G1 cells, in contrast to their S and G2 counterparts, the absence of RIF1 and TP53 is not sufficient to promote HRR and instead promotes an error-prone PARP-EJ. In conclusion, Bart et al. proposed a mechanism in which RIF1 and TP53BP1 mediated the formation of a chromatin barrier around DSBs and promoted NHEJ. RIF1 was recruited along with, but independently of PTIP, and they could cooperate in end protection, but the exact mechanism of this cooperation is not known. As olaparib, a PARP1 inhibitor, is widely used in cancer therapy, RIF1 can be considered a prognostic marker in olaparib-based anti-cancer treatment [106].

To search for the molecular details of RIF1 recruitment to DSB sites, Setiaputra et al. showed that RIF1 had phosphopeptide-binding protein properties directly interacting with three TP53BP1 epitopes [107]. The RIF1-binding sites in TP53BP1 had two phosphorylated residues (S176 and S178) and their mutations abolished RIF1 accumulation into foci induced by ionizing radiation (IR), but the complete abrogation was observed only when an alternative mode of shieldin recruitment to DSB sites was also disabled. The authors concluded that RIF1 used phosphopeptide recognition to promote DSBR and modified shieldin action independently of TP53BP1 binding.

### 5.2. DNA End Resection

As mentioned, DNA end resection is a major effect of DSBR pathway choice. At the initial stage, MRE11 and CtIP are involved, but then it is accomplished by many proteins with a pronounced role of BRCA1 and cyclin-dependent kinase 1 CDK1 [108,109]. CDK1 phosphorylates CtIP at the G1→S transition to initiate end resection and direct DNA repair to HRR [108,110]. However, several other proteins may influence this process, first TP53BP1, which acts antagonistically to BRCA1, suppresses end resection and HRR and promotes NHEJ [111]. On the other hand, BRCA1 may dephosphorylate TP53BP1 to maintain HRR [112]. Therefore, TP53BP1 protects DNA ends against degradation at DSB sites. It does so with other proteins, including RIF1 and Pax transactivation domain-interacting protein (PTIP), which are recruited at DSB in a TP53BP1-dependent manner [113]. Therefore, RIF1 is loaded on the TP53BP1 scaffold to suppress NHEJ and direct DNA repair to HRR. RIF1 recruitment depends on TP53BP1 phosphorylation mediated by ATM [107]. This confirms multifaceted functionality of ATM as it is critical for the end resection process [114]. Exonuclease I is responsible for end resection in HRR and may be recruited to DSB sites by the changes in chromatin structure promoted by the release of RIF1 from the TP53BP1–RIF1 complex and TP53BP1 repositioning [85].

Li et al. showed that DEAD box 1 (DDX1), an RNA helicase involved in DSBR, interacted and colocalized with RIF1 in interphase [115]. DDX1 was recruited to DSB sites in a RIF1-dependent manner and RIF1 depletion abolished DDX1-mediated HRR. The main conclusion of this study was that DDX1 might be responsible for the different consequences of RIF1 depletion, but does not completely have the same consequence as TP53BP1 ablation in restoring HRR defects in BRCA1-deficient cells.

Escribano-Diaz et al. showed that TP53BP1 suppressed BRCA1 accumulation at DSB sites in G1 [116]. ATM-dependent phosphorylation of TP53BP1 recruited RIF1 and TPIP to DSB sites, which are critical effector of TP53BP1 in DSBR. These authors also showed that BRCA1 and CtIP antagonized RIF1 accumulation at DSB sites. PTIP contains the BRCT (BRCA1 C-terminal) domain, facilitating its interaction with TP53BP1 [117]. The relationship between TP53BP1, RIF1, and TPIP is not fully known, and likely complex [118]. Moreover, RIF1 depletion restored end resection and RAD51 (RAD51 recombinase), a critical HRR protein, loading in BRCA1-depleted cells. This work presented RIF1 and BRCA1 as critical elements of DSBR pathway choice to favor NHEJ in G1 phase and HRR in S phase of the cell cycle (Figure 6).

Although RIF1 promotes NHEJ by suppressing DNA end resection, there are several NHEJ variants with distinct mechanisms and proteins involved, so mechanisms of RIF1-mediated NHEJ promotion may be different for different NHEJ variants [119]. This issue was addressed by Han et al. who showed that the TP53BP1-RIF1-Artemis complex promoted alternative pathway of NHEJ (alt-NHEJ) by the retention of Artemis at DSB sites [120]. Earlier, Artemis was shown to be a downstream effector of the TP53BP1-PTIP pathway, promoting processing DNA ends and the canonical NHEJ pathway [121]. These authors also observed that downregulation of RIF1 decreased alt-NHEJ in BRCA2-depleted cells to comparable level as in the case of TP53BP1 depletion. Isobe et al. reported that SCAI (suppressor of cancer cell invasion) bound TP53BP1 phosphorylated at S/TP sites and supported HRR [122]. SCAI gradually replaced RIF1 in the complex with TP53BP1 formed at DNA damage sites. Depletion of SCAI decreased the accumulation of HRR factors, including BRCA1. Therefore, SCAI inhibits RIF1 to activate BRCA1-mediated repair, which may include alt-NHEJ and resection-dependent canonical NHEJ in G1, as well as HRR in S/G2.

### 5.3. Interaction with Shieldin

The protein complex shieldin was identified as a TP53BP1 effector that might be essential for TP53BP1-mediated suppression of end resection [123]. This complex consists of SHLD1 (shieldin complex subunit 1, C20orf196), SHLD2 (FAM35A, family with sequence similarity 35 member A), SHLD3 (CTC-534A2.2), and MAD2L2. Noodermeer et al. showed that shieldin localized at DSB sites in a TP53BP1- and RIF1-dependent manner and its SHLD2 subunit bound ssDNA, protecting it from nucleolytic attack [124]. These authors also showed that SHELD3 directly interacted with RIF1, suggesting that this subunit recruited shieldin to TP53BP1–RIF1 bound to chromatin. These results were independently confirmed by Gao et al. [125]. Findlay et al. identified SHLD2 as an effector of MAD2L2 (mitotic arrest deficient 2 like 2, REV7) in NHEJ and showed that SHLD2-depeltion impaired NHEJ and compromised antibody diversification by class switching recombination in B cells [126]. MAD2L2 was identified as a factor controlling DNA repair at mammalian telomeres [127]. Further experiments showed that MAD2L2 displayed other DDR-related activities, including NHEJ promotion by inhibiting 5′ end resection downstream of RIF1. SHLD2 accumulated at DSBs with dependence on TP53BP1, RIF1, and MAD2L2 and antagonized HRR by blocking DNA end resection. Therefore, SHLD2 may play an important role in the RIF1-mediated decisive mechanism of DSBR choice.

Isobe et al. showed that RIF1, but not shieldin, suppressed the accumulation of CtIP at DSB sites immediately after damage [128]. These authors found that PP1, a known RIF1 effector in DNA replication, suppressed CtIP accumulation and limited the resection by the MRN complex. In conclusion, RIF1 inhibited end resection with PP1 before shieldin action to prevent HRR in early DDR.

Zhao et al. recently identified the RIF1 downstream effector complex ASTE1 (asteroid homolog 1) as a structure-specific DNA endonuclease, specifically resecting DNA 3′ overhangs, acting downstream of the shieldin complex [129]. Therefore, ASTE1 can be a new factor involved in the control of DSBR pathway choice by the TP53BP1-RIF1-shieldin signaling.

Mirman et al. showed that the CST complex consisting of CTC1 (CST telomere replication complex component 1), STN1 (STN1 subunit of CST complex), and TEN1 (TEN1 subunit of CST complex), an accessory factor of the polymerase α-primase complex, was a downstream effector of the TP53BP1-RIF1-shieldin pathway-regulating end resection [130]. Depletion of CST caused an increased end resection similarly to loss of RIF1, TP53BP1, or shieldin. Therefore, CST may assist RIF1, TP53BP1, and shieldin in DSBR regulation (Figure 7).

### 5.4. Epigenetics

Although changes in the epigenetic pattern associate with local changes in chromatin structure, we decided to present the involvement of RIF1 in these effects in separate sections for the sake of clarity.

DSBs trigger a series of ubiquitylation that contributes to the choice of DSBR pathway. This series includes the ubiquitylation of histone H2A by the RNF168 (ring finger protein 168) ligase and the subsequent recruitment of RIF1, which inhibits homologous recombination in G1 cells [131]. The partner and localizer of BRCA2 (BRCA2 DNA repair associated) (PALB2) forms oligomers with BRCA1/2, and this interaction must be tightly regulated to ensure suppression of HRR in G1 and its activation in S and early G2 [132,133]. Signaling pathway involved in the choice between HRR and NHEJ includes a joint action of the E3 ubiquitin ligases RNF8 and RNF168, resulting in the recruitment of the BRCA1-Abraxas-RAP80 (ubiquitin interaction motif containing 1), MERIT40 (BRCA1-A, BRISC and BRCA1 A complex member 1), and TP53BP1 complexes [116]. TP53BP1 is recruited through its effectors RIF1 and MAD2L2 and inhibits end resection.

As HRR requires the presence of an intact template of replicated DNA, cells have evolved mechanisms to block HRR in G1, when after mitosis, a homology partner may be difficult to find (reviewed in [134]). When cells are about to enter S phase, these blocking mechanisms are freed to activate HRR. However, HRR in non-replicated DNA in S phase may promote aberrant DNA structures that may lead to stalling or collapsing of replication fork. Therefore, the choice between NHEJ and HRR may be influenced by the replication status of a DNA fragment—non-replicated vs. replicated. In S phase, NHEJ can process DSBs in pre-replicative DNA and HRR in post-replicative DNA.

Simonetta et al. showed that MAD2L2 (mitotic arrest deficient 2 like 2, REV7) was recruited to DSB sites in H4K20 dimethylated chromatin by forming a complex with TP53BP1 and RIF1 [135]. This complex inhibited the accumulation of BRCA1 at these sites. Simonetta et al. also showed that the choice of DSBR was determined by the replication status of a DNA fragment and epigenetic mechanisms. In non-replicated DNA, the TP53BP1–RIF1–MAD2L2 complex was formed at DSB sites with a subsequent exclusion of BRCA1 at di-methylated lysine 20 of histone 4 (H4K20me2), designating NHEJ as a preferred DSBR pathway. When this DNA fragment was replicated, it caused the dilution of H4K20me2, resulting in TP53BP1–RIF1–MAD2L2 releasing access of BRCA1, end resection, and HRR. Therefore, the H4K20 methylation status in pre-replicative and post-replicative DNA may be an intrinsic mechanism of DSBR pathway choice through the binding/releasing of the TP53BP1–RIF1–MAD2L2 complex.

Drane et al. observed that RIF1 was necessary to separate TP53BP1 from Tudor interacting repair regulator (TIRR) [98]. Interaction of TIRR with TP53BP1 shielded its H4K20me2-binding motif, which is necessary for TP53BP1 recruitment to DSB sites. This work reveals further details of TP53BP1–RIF1 interaction in DSBR, suggesting that the epigenetic pattern of chromatin may play a role. Therefore, local chromatin changes may play a role in the RIF1 signaling in DSBR.

Kumar and Cheok showed that in response to DNA damage, RIF1 was SUMOylated [136]. This process was primarily mediated by the protein inhibitor of activated STAT (PIAS4), a SUMO E3 ligase. Furthermore, PIAS4 downregulation caused impaired RIF1 SUMOylation, defects in the disassembly of DDR-related RIF1 foci, and abolished UHRF1 (ubiquitin like with PHD and ring finger domains 1)-dependent ubiquitination of RIF1, impairing turnover of the RIF1 protein. Kumar and Cheok underlined the role of SUMOylation in disassembly of the RIF1 DDR-related foci, showing that disturbances in this process might lead to DSB induction and concluded that PIAS4 promoted genomic stability by temporal regulation of removal of RIF1 from DSB sites.

### 5.5. Local 3D Chromatin Changes

CSB (ERCC excision repair 6, chromatin remodeling factor) is involved in DSBR, and this involvement is mediated by the interaction with RIF1 through its winged helix domain in S phase [137]. At DSB sites, CBS displaces histones, limiting RIF1 and its effector MAD2L2, but supporting BRCA1 accumulation. As regulation of the DSBR pathway choice by RIF1 is mediated by changes in chromatin, proteins of chromatin remodeling may modulate this regulation, especially when they directly interact with RIF1.

These studies provoke several questions about local changes in 3D chromatin in the vicinity of a DSB, their specificity to replicating DNA, and the involvement of RIF1. These questions have been addressed by “classical” microscopy, but none of them have been fully answered [138,139]. Development of 3D structured illumination microscopy (3D-SIM) and higher-resolution imaging by stimulation depletion (STED) microscopy provided adequate tools to address these problems [140,141]. Sites of DSBs are organized within chromatin in nanodomains containing elementary DDR-related protein foci (nano-foci) [142]. Ochs et al. used live-cell STED microscopy and 3D-SIM and identified four to seven subdomains in a typical TP53BP1 focus [143]. These subdomains were divided into TP53BP1 nano- and microdomains (TP53BP1-NDs and TP53BP1-MD, respectively). RIF1 depletion caused breakage of TP53BP1-MDs into disordered and elongated structures featured by misaligned TP53BP1-NDs. These data suggest that RIF1 forms an autonomous unit with TP53BP1 to stabilize TP53BP1-NDs into an ordered chromatin architecture. At the DSB sites, RIF1 localized to the chromatin between adjacent TP53BP1-NDs and its depletion caused inability of TP53BP1-NDs to organize into higher order structures, such as TP53BP1-MDs. Importantly, disruption of the shieldin complex did not interfere with spatial organization of TP53BP1-NDs. Therefore, RIF1 recruited to DSBs may play a specific role in the stabilization of chromatin mediated by the creation of TP53BP1-NDs. These studies also confirmed that RIF1 cooperated with cohesin to maintain chromatin structure at DSB sites and that RIF1 depletion favored DNA end resection at these sites, as evidenced by BRCA1 expansion. RIF1 was not sufficient to change the chromatin structure, but it was primarily responsible for the stabilization of the structure around a DSB. Finally, the authors speculated that the association of RIF1 with TP53BP1 might have evolved to preserve epigenetic markers hidden in 3D chromatin structure challenged by DSBs.

## 6. RIF1 in Reactivation of Impaired Replication Fork

Activation of the intra-S checkpoint blocks the cell cycle and activates HRR to restart the arrested replication fork (reviewed in [144,145]).

Stalled or collapsed replication fork disturbs RT, and its reactivation is essential for the cell fate, as apart from impaired RT, damaged fork may have a serious delayed consequence, including genomic instability contributing to several pathologies [79]. Therefore, cells evolved pathways to restart damaged replication fork, which can be broadly divided into occurring with or without DSB intermediates [146,147].

An important example illustrating the interconnection between DNA replication and DSBR is the case of replication fork encountering interstrand cross-links (ICLs) (reviewed in [148]). ICLs are induced by an agent covalently bound to the two strands of a DNA molecule, impairing strand separation, a prerequisite in DNA replication, transcription, and recombination. There is not a uniform DNA repair pathway targeting ICLs. The Fanconi anemia network, nucleotide excision repair, HRR, and translesion synthesis (TLS) are involved in ICLs repair with DSBs as an intermediate (reviewed in [149]). Even an accurate repair of ICLs may interfere with RT if the cell is not given enough time for such repair.

Xu et al. showed that RIF1 was part of a multiprotein complex to maintain genomic stability and the BLM complex, containing the BLM helicase [21,150]. Human RIF1 and BLM were recruited to ICLs with the same rate, and the majority of them were colocalized [21] (Figure 8). RIF1 cooperated with BLM to recover the stalled replication fork. The stability of RIF1 depended on the BLM complex. Although there was a direct interaction between the C-terminal domain of RIF1 and the BLM complex, RIF1 was also recruited, although less efficiently, for the stalled replication fork in the absence of BLM. Although previously Buonomo et al. showed that RIF1 colocalized with TP53BP1 and a fraction of BLM, Xu et al. observed that TP53BP1 was not a part of the BLM complex and might be separately recruited to the stalled replication fork [32,151]. In conclusion, RIF1 and BLM worked in the same pathway to prevent damage accumulation and promoted reactivation of stalled replication forks.

The presence of two distinct fork recovery pathways controlled by TP53BP1 and BRCA1 independently of DSBR was shown [151]. Xu et al. showed that TP53BP1 and RIF1 played a NHEJ-independent role in response to replication stress. The absence of TP53BP1-induced hypersensitivity to replication stress was suppressed by the *BRCA1* deletion. TP53BP1 promoted the fast kinetics of fork restart, whereas BRCA1 was involved in slow fork reactivation. The authors claimed that TP53BP1 was dispensable to DSBR at broken replication fork, but BRCA1 promoted MUS81 (MUS81 structure-specific endonuclease subunit)-coupled break-induced recombination (BIR), a subpathway of HRR.

The importance of DSBs and their repair for RT provokes the question about timing of DSBR, especially in meiosis [152]. Moreover, resuming of DNA replication in the case of ICLs repair and some other circumstances is linked with DSBs affecting RT with a strong causal association with genomic instability and cancer (reviewed in [81]).

Mouse RIF1 deficiency induced an increased aphidicolin-evoked replication stress, a defect in the intra-S checkpoint, and an accumulation of DNA damage in S phase [33]. RIF1 localized at sites with stalled replication fork, mostly at the pericentromeric chromatin. A decrease in HRR efficacy was observed in cells with reduced RIF1 levels. This was associated with aggregation of aberrant RAD51. It was shown that with ATR activation and TP53BP1 recruitment, RIF1 localized at a subset of stalled replication forks preferentially in mid-late S phase and regulated HRR to resume replication. RIF1 deletion induced a chromatid-type genome instability. It was hypothesized that RIF1 regulated RAD51-dependent HRR in the regions of the genome that are difficult to replicate, including repeated DNA sequences, which consequently made them prone to replication stress. RIF1 foci at stalled forks colocalized with the BLM RecQ helicase, but not all BLM foci overlapped those of RIF1.

Xu et al. showed that TP53BP1–RIF1 and BRCA1 played different roles than DSBR in the reactivation of replication fork [151]. They observed that only the defect in fork restart but not DSB repair in the TP53BP1-depleted cells was ameliorated by the interruption of BRCA1. The antagonistic roles of TP53BP1 and BRCA1 in replication restart are somehow similar to their roles in DSBR, in which both proteins counteract functions in DSBR. Furthermore, end resection and fork cleavage are the decision steps of DSBR pathway choice and fork restart pathway, respectively. The former is initiated by the CtIP–MRN endonuclease complex, the latter is performed by the MUS–SLX (structure-specific endonuclease subunit) endonuclease complex. The authors hypothesized that TP53BP1 and RIF1 might display a common mechanism in blocking DSBs resection and preventing fork cleavage. This mechanism might include the formation of a higher-order chromatin structure to limit the access of BRCA1-recruited nucleases [98]. This function is apparently in conflict with the ability of RIF1 to recruit the downstream proteins, such as BLM. BRCA1 might disturb the chromatin structures needed for TP53BP1–RIF1 accumulation as BRCA1 displays chromatin-decondensation activity [153]

Matoo et al. showed that myeloid cell leukemia 1 (MCL-1) is essential for the survival of cells exposed to ionizing radiation (IR) [154]. Depletion of MCL-1 in IR-exposed hematopoietic stem cells increased genomic instability and the number of residual TP53BP1 and RIF1 foci and decreased formation of foci related to HRR. An analogous effect was observed in MCL-1-depleted cells with stalled replication fork induced by hydroxyurea. In conclusion, MCL-1 depletion increases TP53BP1 and RIF1 colocalization and inhibits BRCA1 recruitment and HRR needed for repairing DSBs caused by IR or resolution of stalled replication fork.

To look for a mechanism linking the involvement of RIF1 in RT and DSBR, Saito et al. exposed HeLa cells to high doses of IR [155]. They observed that IR suppressed HRR and this effect was mediated by RIF1. IR did not affect NHEJ. At low IR doses, BRCA1 inhibited RIF1 in S/G2 phase. At high IR doses, RIF1 dephophorylated the MCM helicase, resulting in suppression of initiation of DNA replication. Dephosphorylation of MCM results in inhibition of both end resection and HRR even independently of RIF1 [156]. The authors concluded that MCM might be a possible link between DNA replication and HRR, and RIF1 controlled the suppression of both initiation of replication and HRR at high IR doses.

In summary, RIF may be involved in RT, DSBR, and fork reactivation by mechanisms that may partly overlap, but the exact determination of the extent of that overlapping requires further research. In general, RIF1 may regulate RT by changes in the 3D structure of chromatin; binding of G4 quadruplexes, which is common in telomeres; and activation deactivation of proteins involved in the initiation of DNA replication, including the MCM helicase (Figure 9). Impaired functions of replication fork, resulting from its stalling, slowing, or collapsing, strongly affect RT, and RIF1 may reactivate the impaired fork through mechanisms that can either be related to or independent of its functions in DSBR. RIF1 may play an important role in the choice of DSBR pathway through its association with phosphorylated TP53BP1 and suppression of DNA end resection in G1 phase of the cell cycle, which is required in HRR, but not needed in NHEJ. On the other hand, lack of such action in S/G2 phase of the cell cycle promotes HRR.

## 7. Conclusions and Perspectives

Replication timing is an essential program to ensure timing, complete and accurate replication of the human genome, and adjustment of DNA replication to other ongoing cellular processes. RIF1 is a master regulator of RT in human cells, but it is a multifaceted protein whose essential functions may be unknown.

Apart from RT regulation, RIF1 is involved in DSBR and reactivation of damaged replication fork. The involvement of RIF1 in DSBR is mainly associated with the role of RIF1 in the repair choice, but some studies suggest that it can also be involved in the executional phase of DNA repair. There is no doubt that these functions may overlap, but the extent of overlapping and shared mechanisms of these RIF1 activities are poorly known. Therefore, future research addressing further details of common regulation of RT and DSBR and reactivation of damaged replication fork are needed. Moreover, as DNA end resection is a key element in SSA, as the third main DSBR pathway and the main mechanism underlying DSBR pathway choice by RIF1, the involvement of RIF1 in SSA regulation is justified to study.

The data obtained so far suggest that the structure of chromatin may be essential for all known aspects of RIF1 activity. It is not surprising as changes in 3D structure of chromatin are inextricably associated with RT, DSBR, and fork reactivation. As shown, the development of new imaging methods, including high-resolution microscopy, has a high potential to reveal new details of the involvement of RIF1 in 3D chromatin changes associated with its activities.

As yeast Rif1 phosphorylation sites were shown to play an important role in vital yeast function, it is justified to explore the role of corresponding sites of human RIF1 [24].

When impaired, all three phenomena: RT, DSBR, and fork reactivation may contribute to genomic instability, which is typical for most, if not all cancer cells. Therefore, RIF1 has potential in cancer diagnosis and therapy [35,36,37]. Surely, the immediate question in this regard is whether *RIF1* plays a role of oncogene or tumor suppressor gene and the answer is almost equally immediate as RIF1 is an effector of TP53BP1, which, in turn, positively regulates TP53, a tumor suppressor [157,158]. However, the role of TP53–TP53BP1 interaction in cancer transformation is not completely clear, so the corresponding role of RIF1 should be further explored [159].

When a cell encounters lesion slowing, stalling, or collapsing of replication fork, it may decide to bypass that lesion, which may cause mutations contributing to pathological consequences [144]. A potential role of RIF1 in this process has not been explored yet.

Distinct DSBR functions of RIF1 in replicated and non-replicated DNA are another key element to better understand its diverse actions.

In summary, RIF1 displays several activities, which are mainly associated with its involvement in RT and DSBR and reactivation of damaged replication fork. This involvement is underlined by the interaction with many other proteins, including initiators of DNA replication, cell cycle regulators, DDR proteins, modifiers of epigenetic pattern, and chromatin organization. This makes RIF1 an important element in physiology and pathology with potential in the therapy of many diseases.

## Figures and Tables

**Figure 1 ijms-22-11440-f001:**
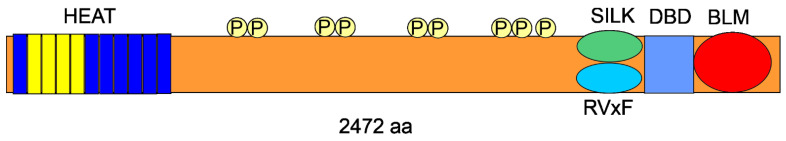
Schematic linear representation of the human replication timing regulatory protein 1 (RIF1). It has 2472 aa and contains conserved domains: HEAT (huntingtin, elongation factor 3, a subunit of protein phosphatase 2A and TOR)/Armadillo-like helical repeats) with its core conserved subdomains in yellow, SILK-RVxF domain, which is a docking motif for protein phosphatase 1 (PP1), DNA binding domain (DBD) and the BLM interacting domain involved in recruiting the BLM helicase to stalled replication fork. At least three clusters of sites of phosphorylation, essential for RIF1 activity, can be considered, and their approximate positions are indicated with circled “P”s (https://www.phosphosite.org/proteinAction.action?id=6427&showAllSites=true, accessed on 19 October 2021). All these domains are only presented with approximate locations and scale.

**Figure 2 ijms-22-11440-f002:**
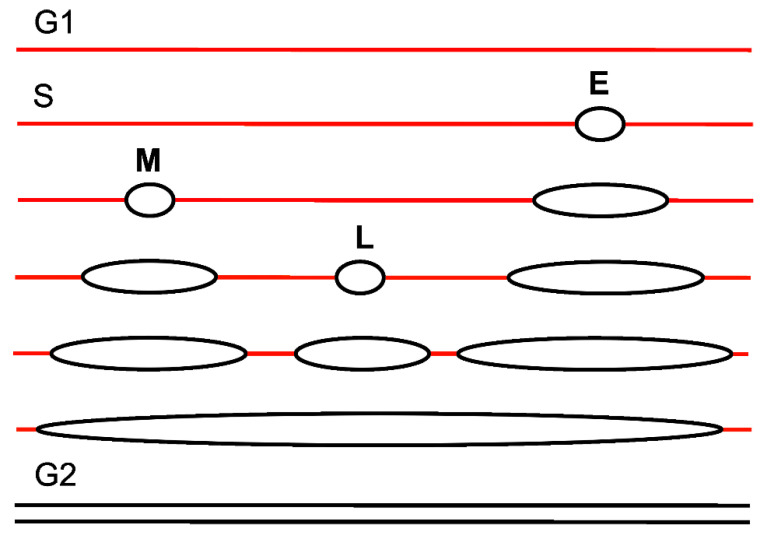
Simplified idea of replication timing. In this scheme, a red or black single line represents a parent or daughter double-stranded DNA, respectively. In late G1, a DNA molecule is ready to replicate and enter S phase. In S, some origins of replication are early activated (fired) and represented here for just a single origin (E). Then some other origins are fired, which are also represented by a single origin (M) and replication initiated in E prolongs. Next, further origins are fired (L), when replications are initiated at E and M progress. If the replication is not disturbed, the cell enters G2 phase with completely replicated DNA. Such an order of events spares energy and decreases the chance of steric conflict between proteins involved in replication initiation.

**Figure 3 ijms-22-11440-f003:**
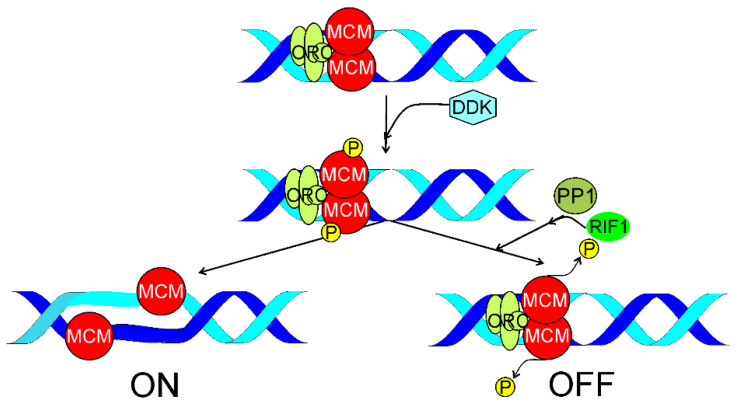
Replication timing regulatory factor 1 (RIF1) may be involved in replication timing (RT) by suppressing the activation of the origin of replication (ori). Origin recognition complex (ORC) and CDC6 (cell division cycle 6, not shown) recruit two MCM2-7 (minichromosome maintenance complex components 2–7, MCM)–CDT1 (chromatin licensing and DNA replication factor 1, not shown) heptamers onto an origin of replication. Two hexamers of MCM form a ring that encircles DNA duplex at ori and are phosphorylated by DBF4 (DBF4 zinc finger)-dependent kinase (DDK). Further recruitment of DNA polymerase α/primase and other proteins of the replication machinery results in DNA synthesis (ON). RIF1 can guide PP1 (protein phosphatase 1) to ori to reverse DDK-mediated phosphorylation of MCM to inhibit initiation of DNA replication (OFF) and change the RT program.

**Figure 4 ijms-22-11440-f004:**
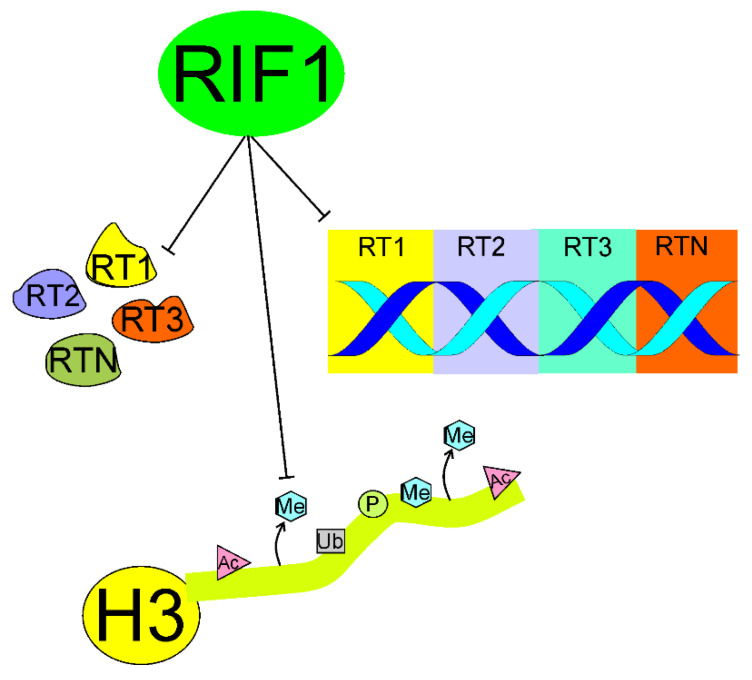
Replication timing regulatory factor 1 (RIF1) may coordinate the replication timing (RT) in individual cells preventing their heterogeneity, which is presented as different RT programs R1…RN operating in different cells (left). RIF1 may prevent depletion of some specific histone marks, suppressing chromatin remodeling-induced disturbances in RT (middle). Histones are presented here by a single H3 with its fold domain (yellow circle) and N-termina tail, which is targeted by histone-modifying proteins; C-terminal tail is not shown; presented modification acetylation (Ac), methylation (Me), phosphorylation (P), and ubiquitinylation (Ub) are for illustrative purposes only and do not reflect any connection to a specific RT program. RIF1 may also suppress compartmentalization of the genome into regions with different RT programs (right). The genome is represented here as a single DNA molecule.

**Figure 5 ijms-22-11440-f005:**
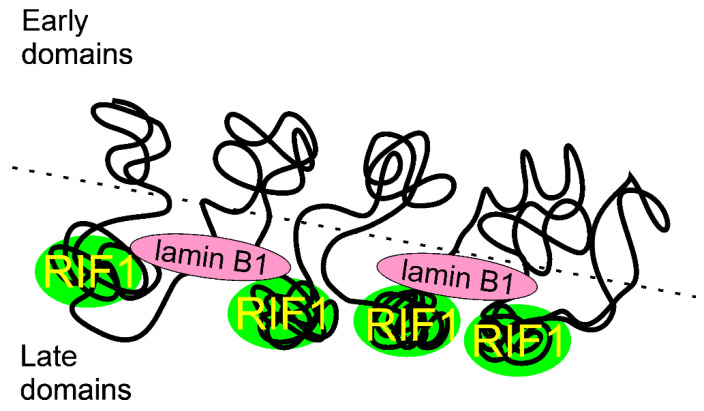
Replication timing regulatory factor 1 (RIF1) may control replication timing by marking late replicating regions in chromatin and constrains inter-domain contacts in G1 through organization of the nuclear architecture and interaction with lamin B1. The interaction of RIF1 with chromatin and lamin B1 is presented in a simplified way as RIF1 and lamin B1 likely coat chromatin fibers that are to be replicated later.

**Figure 6 ijms-22-11440-f006:**
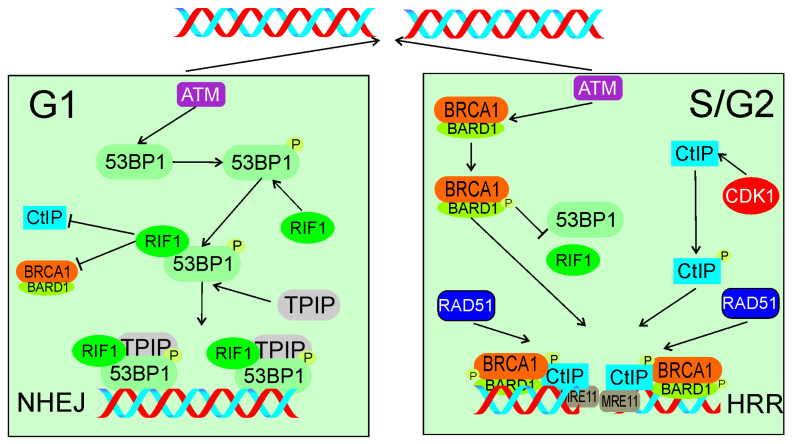
In response to DSBs in G1 (left panel), ATM phosphorylates TP53BP1 (53BP1), which associates with its critical effectors RIF1 and TPIP. These two proteins form a complex to block DNA end resection activity of CtIP and BRCA1/BARD1 and support NHEJ. In S/G2 (right panel), ATM phosphorylates BARD1 and CDK1 phosphorylates CtIP. Phosphorylated BRCA1/BARD1 prevents TP53BP1 binding by RIF1 and recruits phosphorylated CtIP and MRE11 to DSB sites to initiate end resection with subsequent recruitment of HRR machinery represented here by RAD51. Phosphorylation is marked by only single phosphate residue (P). The events presented in the right panel may also lead to microhomology-mediated end joining. All abbreviations are defined in the main text.

**Figure 7 ijms-22-11440-f007:**
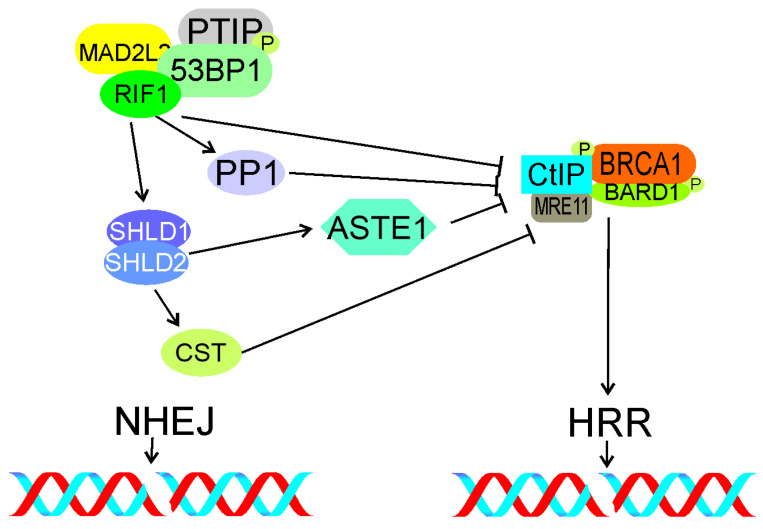
The TP53BP1 (53BP1)-RIF1-shieldin signaling with the choice of DSBR pathway in G1. RIF1 with phosphorylated TP53BP1 and assisted by MAD2L2 and TPIP, inhibits accumulation of phosphorylated CtIP and BRCA1 at DSB sites, preventing their interaction with MRE11 and end resection, and as a consequence, inhibits HRR. Shieldin, represented here by the SDHLD1–SHLD2 complex, promotes NHEJ by interaction with NHEJ proteins (not shown). It may be also involved in CtIP/BRCA1 inhibition (not shown). The CST complex inhibits end resection and HRR downstream of shieldin, so does the ASTE complex. Phosphorylation is marked by only single phosphate residue (P). All abbreviations are defined in the main text.

**Figure 8 ijms-22-11440-f008:**
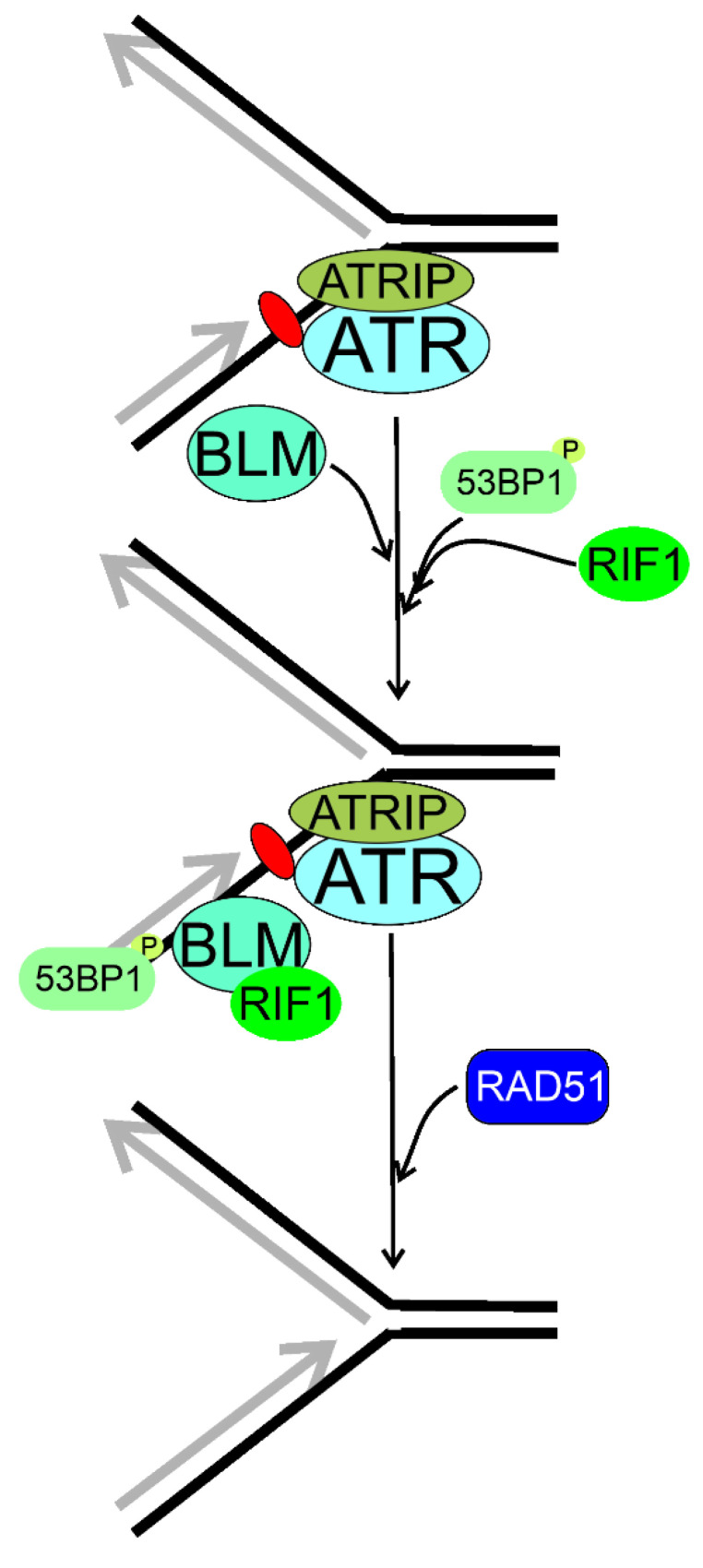
Replication timing regulatory factor 1 (RIF1) may be involved in the reactivation of stalled replication fork, which is recognized by ATR (ATM-and-RAD3-related) acting in concert with its partner, ATRIP (ATR interacting protein). RIF1 associates with phosphorylated (P) T53BP1 (53BP1, tumor protein p53 binding protein) and colocalizes with the BLM RecQ helicase foci. Then, the homologous recombination repair machinery symbolized here by RAD51 is recruited to recover the stalled fork. Here, stalling of the replication fork results from a single DNA lesion (purple oval) in the lagging strand, but many other DNA damages may induce such effects. Moreover, RIF1 may stabilize the replication fork and prevent its collapse or reversal, but the exact mechanism of this RIF1 activity is not completely known. T53BP1 might not be of the RIF1–BLM complex and might be separately recruited to the stalled replication fork.

**Figure 9 ijms-22-11440-f009:**
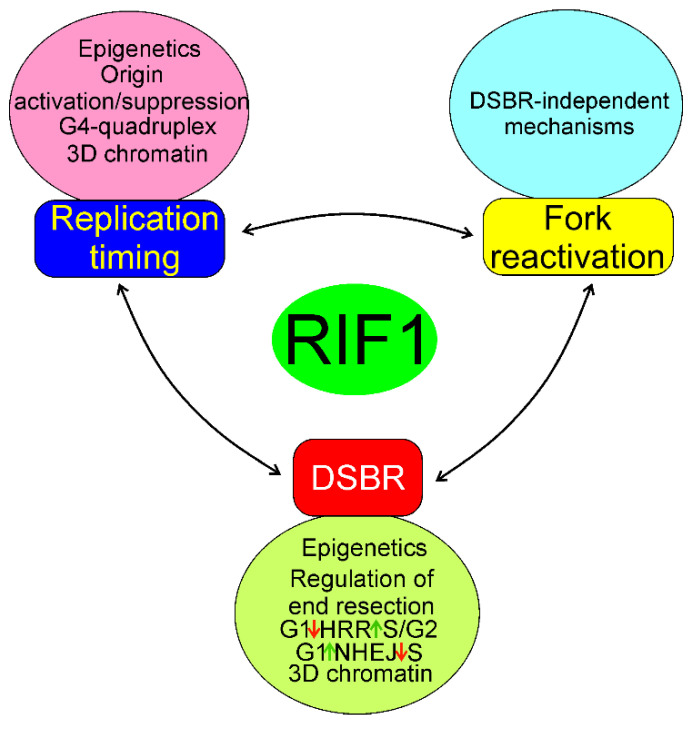
Mechanisms of the involvement of RIF1 in the regulation of RT, fork reactivation, and DSBR. Epigenetics here represent changes in the cellular epigenetic pattern associated with RIF1 action, and they, along with concomitant changes in chromatin structure, are essential for all processes of RIF1. All three effects: RT, fork reactivation, and DSBR are mutually dependent, but the extent and details of this dependency are to be established. All abbreviations are defined in the main text.

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
