# Peer review of "RIF1 Links Replication Timing with Fork Reactivation and DNA Double-Strand Break Repair"

_ijms, 2021, doi:10.3390/ijms222111440_

Round 1

Reviewer 1 Report

The authors well summarize research results, which have been published by many researchers about RIF1 functions as RT, DSBR and fork reactivation thus far. Therefore, the review article should be useful for researchers working in the field. However, it was not easy for me to understand the manuscript, because many abbreviations are used in the manuscript, a few figures for helping development of an understanding are not given and some mistakes can be found in the manuscript. So, I would like to recommend that the authors revise the manuscript according to my comments described below, if my indications are correct.

Major comments:

  1. Figures helping development of an understanding should be given in the two sections of “4. RIF1 in Replication Timing” and “6. RIF1 in Reactivation of Impaired Replication Fork”.  

Minor comments

  1. Font size of the sentence on line 199 should be revised.
  2. Three “PTIP”s in the Figure 3, left panel, should be “TPIP”.
  3. Arrow drawn from ATM to CtIP in the Figure 3, right panel, should be removed.
  4. “(right panel) should be added ”after S/G2 on line 452 of the legend of Figure 3.
  5. Line 4 from the top of the legend of Figure 3: “BRCA1”should be “BARD1”.
  6. Line 2 from the bottom of Figure 4: a phrase “so das the ASTE complex” should be something wrong.
  7. The word “bresulting”on line 557 should be rewritten as “resulting”.
  8. The word “SUMOlyation” crossing from line 621 to 622 should be “SUMOylation”.
  9. “)”should be added after “[143,144]” on line 633, as “(reviewed in [143,144])”.
  10. The abbreviation “(ICs)”on line 640 after “interstrand cross-links” should be “(ICLs)”.
  11. “DSBR” on line 1 from the top in light blue circle of Figure 5 should be removed.
  12. Double spaces on lines 103, 455, 508 and 767 should be revised to single space.

Author Response

The authors well summarize research results, which have been published by many researchers about RIF1 functions as RT, DSBR and fork reactivation thus far. Therefore, the review article should be useful for researchers working in the field. However, it was not easy for me to understand the manuscript, because many abbreviations are used in the manuscript, a few figures for helping development of an understanding are not given and some mistakes can be found in the manuscript. So, I would like to recommend that the authors revise the manuscript according to my comments described below, if my indications are correct.

Major comments:

Comment: Figures helping development of an understanding should be given in the two sections of “4. RIF1 in Replication Timing” and “6. RIF1 in Reactivation of Impaired Replication Fork”.

Answer: We have added four figures – 3 to section 4(Figs. 3-5) and 1 to section 6.

Minor comments

Comment:  Font size of the sentence on line 199 should be revised.

Answer: We have changed font size to 10 points.

Comment: Three “PTIP”s in the Figure 3, left panel, should be “TPIP”.

Answer: We have corrected this typo.

Comment: Arrow drawn from ATM to CtIP in the Figure 3, right panel, should be removed.

Answer: We have done so.

Comment: “(right panel) should be added ”after S/G2 on line 452 of the legend of Figure 3.

Answer: We have done so.

Comments: Line 4 from the top of the legend of Figure 3: “BRCA1”should be “BARD1”.

Line 2 from the bottom of Figure 4: a phrase “so das the ASTE complex” should be something wrong.

The word “bresulting”on line 557 should be rewritten as “resulting”.

The word “SUMOlyation” crossing from line 621 to 622 should be “SUMOylation”.

 “)”should be added after “[143,144]” on line 633, as “(reviewed in [143,144])”.

 The abbreviation “(ICs)”on line 640 after “interstrand cross-links” should be “(ICLs)”.

 “DSBR” on line 1 from the top in light blue circle of Figure 5 should be removed.

    Double spaces on lines 103, 455, 508 and 767 should be revised to single space.

Answer: We have addressed all these comments in the revised version of the manuscript.

Reviewer 2 Report

The review entitled “RIF1 Links Replication Timing with Fork Reactivation and DNA Double-Strand Break Repair” by Blasiak et al. summarizes the biological functions of Replication timing regulatory factor 1 (RIF1), focusing specifically on its role in replication timing, interconnections with DNA repair pathways and replication fork reactivation. Additionally, the authors discuss the role of this important protein in other aspects of cellular metabolism, such as organizing the 3D structure of chromatin, interaction with G4-quadruplexes, or its effect on the epigenome.

Overall, the review provides the comprehensive overview of recent findings on the RIF1 protein. The review is of a good quality, it reads quite well and might be accepted after some corrections.

Comments and Recommendations:

Lanes 37-38: Should read “This program ensures that all chromosomes are replicated once and only once per cell cycle.”

Lane 43: Delete the “DSBs stall or collapse replication fork.” It´s redundant.

Lanes 87-89: Should read “The maximal expression of the RIF1 gene is observed in late G2/S phase, suggesting that RIF1 may perform its most important functions in this phase of the cell cycle [20].”

Lane 100: Should read “…and molecular weight of 274,466 kDa.”

Lanes 105-106: The phosphorylation sites of RIF1 in various organisms (human, yeasts, etc.) should be listed here and their biological significance should be briefly discussed (check 10.1016/j.dnarep.2018.03.001 and other papers).

Lane 109: Should read “Mouse RIF1 CTD contains three CI-CIII segments…”

Lane 274: Authors stated that “Both cohesin or its binding factor and the CCCTC-binding factor disordered TADs, with no effect on RT domains [69-71].” Specify, please, which binding factor of cohesion this concerns.

Lane 365: More details on how TRF1 regulates telomere length and function should be provided here.

Lane 420: Specify, please, which residues within the RIF1-binding sites in TP53BP1 are phosphorylated.

Lanes 465-467: Should read “This work presented RIF1 and BRCA1 as critical elements of DSBR pathway choice to favor NHEJ in G1 phase and HRR in S phase of the cell cycle (Figure 3).”

Lanes 531-541 and 549-560: These paragraphs are identical. Please, make correction.

Lanes 579-587 and 619-630: Again, these sections are almost identical.

Lane 723: Should read “In general, RIF1 may regulate RT by changes in the 3D structure…”

Lanes 729-732: Should read “RIF1 may play an important role in the choice of DSBR pathway through its association with phosphorylated TP53BP1 and suppression of DNA end resection in G1 phase of the cell cycle, required in HRR, but not needed in NHEJ. On the other hand, lack of such action in S/G2 phase of the cell cycle promotes HRR.

To make the review more comprehensive, the authors may consider to add the Table summarizing all discussed RIF1 interactors, including their biological functions.

Author Response

The review entitled “RIF1 Links Replication Timing with Fork Reactivation and DNA Double-Strand Break Repair” by Blasiak et al. summarizes the biological functions of Replication timing regulatory factor 1 (RIF1), focusing specifically on its role in replication timing, interconnections with DNA repair pathways and replication fork reactivation. Additionally, the authors discuss the role of this important protein in other aspects of cellular metabolism, such as organizing the 3D structure of chromatin, interaction with G4-quadruplexes, or its effect on the epigenome.

Overall, the review provides the comprehensive overview of recent findings on the RIF1 protein. The review is of a good quality, it reads quite well and might be accepted after some corrections.

Comments and Recommendations:

Comments: Lanes 37-38: Should read “This program ensures that all chromosomes are replicated once and only once per cell cycle.”

Lane 43: Delete the “DSBs stall or collapse replication fork.” It´s redundant.

Lanes 87-89: Should read “The maximal expression of the RIF1 gene is observed in late G2/S phase, suggesting that RIF1 may perform its most important functions in this phase of the cell cycle [20].”

Lane 100: Should read “…and molecular weight of 274,466 kDa.”

Answer: These all comments were addressed in the revised version of the manuscript.

Comment: Lanes 105-106: The phosphorylation sites of RIF1 in various organisms (human, yeasts, etc.) should be listed here and their biological significance should be briefly discussed (check 10.1016/j.dnarep.2018.03.001 and other papers).

Answer: The biological significance of the phosphorylation sites of RIF1 is best known in yeast and I do not know any comparable study in humans or other mammals. As we do not like to write too much about yeast Rif1, especially about research not directly to humans, we have only added the following fragment to the corresponding part in the manuscript:

“Wang et al. showed that phosphorylation of yeast Rif1 had both positive and negative effects on telomere length regulation [24]. These authors suggested that as all yeast Rif 1 orthologues, including human one, had one or more conserved serine or threonine followed by glutamine (SQ/TQ) cluster domains targeted by the ATM family kinases, the results they obtained might have a more general significance and implications for the regulation of RIF1 functions in human and other organisms.“

and added the following sentence to the concluding section:

“As yeast Rif1 phosphorylation sites were shown to play an important role in vital yeast function, it is justified to explore the role of corresponding sites of human RIF1 [24].”

with the new reference:

  1. Wang, J.; Zhang, H.; Al Shibar, M.; Willard, B.; Ray, A.; Runge, K.W. Rif1 phosphorylation site analysis in telomere length regulation and the response to damaged telomeres. DNA repair 2018, 65, 26-33, doi:10.1016/j.dnarep.2018.03.001.

Comment: Lane 109: Should read “Mouse RIF1 CTD contains three CI-CIII segments…”

Answer: We have corrected that typo.

Comment: Lane 274: Authors stated that “Both cohesin or its binding factor and the CCCTC-binding factor disordered TADs, with no effect on RT domains [69-71].” Specify, please, which binding factor of cohesion this concerns.

Answer: We have changed the sentence:

“Both cohesin or its binding factor and the CCCTC-binding factor disordered TADs, with no effect on RT domains [69-71].”

into:

“Both cohesin or its loading factor NIPBL (NIPBL cohesin loading factor) and the CCCTC-binding factor disordered TADs, with no effect on RT domains [69-71].”

Comment: Lane 365: More details on how TRF1 regulates telomere length and function should be provided here.

We have added the following fragment (line 365 in the original manuscript):

“RIF1 binds at dysfunctional mammalian telomeres to recover them with its DSBR-related activity rather than telomere-specific functions. However, Adams and McLaren showed that mouse RIF1 highly expressed in primordial germ cells and embryo-derived pluripotent stem cell lines directly interacted with the telomere-associated protein TRF2 and could be crosslinked to telomeric repeat DNA in mouse embryonic stem cells [93]. These authors also showed that RIF1 assisted to limit the expression of the ZSCAN4 (zinc-finger and SCAN domain-containing 4) gene, whose product supports a recombination-related mechanism of telomere-elongation. Therefore, RIF1 can be involved in DSBR-independent telomere maintenance in the germline and early mouse development. Further works are needed to definitely determine the role of mammalian RIF1 in telomere maintenance and clarify the reason of difference between that role and its yeast counterpart.”

with the new reference

  1. Adams, I.R.; McLaren, A. Identification and characterisation of mRif1: a mouse telomere-associated protein highly expressed in germ cells and embryo-derived pluripotent stem cells. Dev Dyn 2004, 229, 733-744, doi:10.1002/dvdy.10471.

Comment: Lane 420: Specify, please, which residues within the RIF1-binding sites in TP53BP1 are phosphorylated.

Answer: We have changed the sentence:

“The RIF1-binding sites in TP53BP1 had two phosphorylated residues and their mutations abolished RIF1 accumulation into foci induced by ionizing radiation (IR), but the complete abrogation was observed only when an alternative mode of shieldin recruitment to DSB sites was also disabled.”

into:

“The RIF1-binding sites in TP53BP1 had two phosphorylated residues (S176 and S178) and their mutations abolished RIF1 accumulation into foci induced by ionizing radiation (IR), but the complete abrogation was observed only when an alternative mode of shieldin recruitment to DSB sites was also disabled.”

Comment: Lanes 465-467: Should read “This work presented RIF1 and BRCA1 as critical elements of DSBR pathway choice to favor NHEJ in G1 phase and HRR in S phase of the cell cycle (Figure 3).”

Answer: We have corrected that sentence accordingly.

Comments: Lanes 531-541 and 549-560: These paragraphs are identical. Please, make correction.

Lanes 579-587 and 619-630: Again, these sections are almost identical.

Answer: We have removed unnecessary repeated fragments.

Comments: Lane 723: Should read “In general, RIF1 may regulate RT by changes in the 3D structure…”

Lanes 729-732: Should read “RIF1 may play an important role in the choice of DSBR pathway through its association with phosphorylated TP53BP1 and suppression of DNA end resection in G1 phase of the cell cycle, required in HRR, but not needed in NHEJ. On the other hand, lack of such action in S/G2 phase of the cell cycle promotes HRR.

Answer: We have corrected these sentences.

Comment: To make the review more comprehensive, the authors may consider to add the Table summarizing all discussed RIF1 interactors, including their biological functions.

Answer: We think that addition of such a table would largely repeat information contained in the figure in the concluding section (Figure 5 in the original manuscript, Figure 9 in its revised version) and in consequence – overload the manuscript.